# Continuous Based Direct Ink Write for Tubular Cardiovascular Medical Devices

**DOI:** 10.3390/polym13010077

**Published:** 2020-12-28

**Authors:** Enric Casanova-Batlle, Antonio J. Guerra, Joaquim Ciurana

**Affiliations:** 1Grup de Recerca en Enginyeria Producte Procès i Producció (GREP), Universitat de Girona, 17003 Girona, Spain; enric.casanova@udg.edu; 2EURECAT, Centre Tecnològic de Catalunya, 08005 Barcelona, Spain; antonio.guerra@eurecat.org

**Keywords:** Direct Ink Write, stent, vascular graft

## Abstract

Bioresorbable cardiovascular applications are increasing in demand as fixed medical devices cause episodes of late restenosis. The autologous treatment is, so far, the gold standard for vascular grafts due to the similarities to the replaced tissue. Thus, the possibility of customizing each application to its end user is ideal for treating pathologies within a dynamic system that receives constant stimuli, such as the cardiovascular system. Direct Ink Writing (DIW) is increasingly utilized for biomedical purposes because it can create composite bioinks by combining polymers and materials from other domains to create DIW-printable materials that provide characteristics of interest, such as anticoagulation, mechanical resistance, or radiopacity. In addition, bioinks can be tailored to encounter the optimal rheological properties for the DIW purpose. This review delves into a novel emerging field of cardiovascular medical applications, where this technology is applied in the tubular 3D printing approach. Cardiovascular stents and vascular grafts manufactured with this new technology are reviewed. The advantages and limitations of blending inks with cells, composite materials, or drugs are highlighted. Furthermore, the printing parameters and the different possibilities of designing these medical applications have been explored.

## 1. Introduction

Additive manufacturing methods are gaining prominence due to the ability to customize the shape of the carried out applications in an inexpensive manner. That is why many teams are researching innovative technologies to extrude different types of material. The most common, Fused Deposition Modeling (FDM), was developed by S. Scott Crump [1] and is also known as Fused Filament Fabrication or Free Form Fabrication (FFF). This method employs the viscoelastic characteristic of shear thinning behavior of non-Newtonian thermoplastic materials, which easily flow under shear stress and quickly recover upon deposition, to produce a continuous filament during the printing process. Thus, the molten material is subjected to mechanical pressure, leading to shear deformation as it passes through the nozzle system, high shear forces. Then, it is collected on the deposition bed where it is cooled following a precise pattern. Once deposited, the shear force is no longer present, it cools and maintains its conformation due to its rapid cooling and low specific heat capacity. These phenomena rapidly increase its viscosity due to the shear thinning feature. Therefore, the material shows higher resistance to deformation. In addition to the property of shear thinning, to use this technique, the material must encompass the ability to make a filament with the desired printed material, as well as heat the material to the melting point and have a low specific heat capacity [2]. Hence, there is a limited stock of materials that fulfills these demands and, therefore, can be applied to that technique. However, there are many materials that possess the ability of shear thinning, in addition to other properties [3,4]. Sandia National Laboratories, in 1996, defined another way to deposit polymers, which they called robocasting [5]. Today, this technique is well known as Direct Ink Writing (DIW). This technology is based on the extrusion of viscoelastic pastes, also referred as ’inks’, through a nozzle (50 μm to 1.25 mm [6,7]), forming a 3D structure. The shear thinning property in DIW is even more crucial than in FDM, since no heat treatment is conducted, therefore the material relies exclusively on this feature to harden after being deposited on the bed. The shear stress to create the flow is generally generated by the action of a pneumatic or mechanical piston or an Archimedes screw that is connected to the syringe barrel containing the ink [8]. Consecutively, this mechanism is coupled to a nozzle that is mounted on a 3-axis carriage that precisely controls the placement of the deposited material. That technology offers a whole new range of polymeric materials in which additive manufacturing can be carried out. Employing this technique, materials like epoxy resins [9,10], preceramic polymers [7,11,12,13,14], natural polymers [3,15,16], or conductive composite polymers [17,18,19,20,21] can be tailored as convenience for biomedical applications. Furthermore, it is the preferred technique for printing living cells by mixing them in hydrogels due to the high survival rate [22,23,24,25,26], in addition to the biocompatible water-rich environment that simulates the extracellular matrix (ECM) [27]. DIW has several advantages for biomedical applications. This new additive manufacturing approach is very interesting for the elaboration of more complex tissues that require a precise control of cell deposition for their subsequent interactions and development. Being able to customize or specifically tailored each application for its final user during the fabrication process adds an added-value that cannot be provided by industrialized manufacturing processes [19,28,29]. DIW has the ability to build any desired shape or construction with any material that encompasses the shear thinning property. In addition, a multimaterial ink can be blended, providing characteristics of interest of both components and creating very attractive biomedical engineering devices [6,18]. DIW is gaining more and more presence within the cardiovascular tubular medical treatments, as the implicit viscoelastic property required to conduct the technique is ideal for the deposition of the ink on a tubular rotating bed. That is why this review delves into this novel field which might embrace solutions to very common problems today.

## 2. Polymeric Material Features for Cardiovascular DIW

Designing devices for the cardiovascular system is not an easy task. The blood stream is a very inhospitable environment where any invading body is immediately attacked by the immune system. Therefore, the materials used to design the medical devices that are intended to be introduced in the blood vessels must meet the highest standards of biocompatibility. Additionally, the DIW technique arrives in the cardiovascular field to address differently the issue that has been approached permanently in recent years, thus proposing bioresorbable materials that possess sufficiently high mechanical properties to withstand the working conditions without causing blood clots. Moreover, to comply with the DIW requirements, the deposited solution should also fulfill certain rheological characteristics.

### 2.1. Rheological Properties

Printed polymers require the shear thinning viscosity feature, relativity low viscosity under stress, and good shape retention capacity when it is extruded on the tubular bed. Inks with viscosities less than 0.3 Pa·s under non-shear stress conditions are not capable of maintaining the 3D pattern [22,30]. Higher viscosities (>100 Pa·s) induce nozzle clotting, making printing unfeasible. Therefore, viscosities between 0.3 Pa·s and 100 Pa·s are ideal [31]. Additionally, rapid self-healing ability after deposition is desired. This feature ensures that the liquid-solid transition is fast enough so that the ink does not scatter before it retains its final shape. This is encountered with a large difference between the storage modulus (G’) and the loss modulus (G’’) [32]. These parameters represent how elastic and viscous the ink is. They can also be represented by the complex modulus (G*), which is the vector contributions of both modules. Finally, the angle between the complex modulus and the storage modulus is known as the phase angle (δ). If this angle is close to 0 rad, the material exhibits solid-like behavior, whereas, if it is close to π rad, it presents liquid-like properties. Therefore, the capability to modulate these features during the method plays an important role in the printing process. During the high shear forces of the extrusion phase, it is preferred that the G’’ be greater than the G’ in order for the ink to flow through the nozzle. Contrastingly, G’ must be greater than G’’ in the deposition phase, low shear, or non-shear stress, as it should be large enough to retain the printed characteristics. Plus, the material may need to sustain other layers of materials on top. Therefore, G’ modulus must be sufficiently high to support the load of the 3D printed application. Thus, the greater the difference between these modules, at low-shear stress, the faster is the transition between solid-fluid states and, therefore, the more accurate the system can print. Alternatively, materials with the ability to change from a more solid state to a more liquid state are also called yield-stress materials [33]. The materials in which the yield stress point, the point at which the applied stress deforms the solid with a liquid behavior, is encountered when the ink passes through the nozzle may be potentially suitable for DIW. The yield stress point is at the intersection between G’ and G’’ as a function of the shear stress. This characteristic is found in polymers because their molecular structure is based on cross-linked physical networks between polymer chains. These bonds are broken by the applied shear forces, but, once the shear stress is eliminated, these interactions are restructured.

### 2.2. Storage Modulus and Loss Modulus Effect on Printing Resolution

Table 1 reviews various ink blends used in medical applications. The relationship between G’ and G” is shown to have a great effect on the print quality. A compromise between these parameters should be desired for DIW. To provide sufficient rapid self-healing ability, ideally, the G’ should be at least 200 Pa greater than the yield stress point for robocasting purposes [34]. Moreover, a G’’/G’ ratio of less than 0.8 is recommended for good printability as it ensures a rapid liquid-solid transition and typically a G’ large enough to sustain the structure [35]. As shown in Table 1, Cheng et al. [15] formulated an acrylamide-based bioink. They added sodium alginate and calcium chloride in different concentrations to craft the shear thinning property of the inks. The ink with the highest concentration of both reagents reported a maximum difference of 2000 Pa between these parameters, with the G’’/G’ ratio ∼0.5. The resulting printed material efficiently maintained the shape of the deposition pattern. However, in the same way, the ink with the lowest concentration of reagents also had a G’’/G’ ratio of ∼0.5. Nevertheless, a difference of 100 Pa was found between the modulus. Although the resulting ink was viscous enough for DIW, it maintained the shape of the pattern, the ink scattered approximately twice as much than the highest concentrations ink [15]. In contrast, the ink containing fumed silica had a better ability to retain shape after deposition than the equivalent ink without this component since the reconstruction of hydrogen bonds between silica particles is faster than the misalignment of polymeric chains, as in Table 1 [36]. Both inks had a G’-G’’ difference of 5000 Pa. However, the ink containing fumed silica had a lower G’’/G’ ratio. This is in agreement with Li et al. [35], who state that the lower the G’’/G’ ratio, the greater the ability to retain shape after ink deposition. Nevertheless, the fumed silica-free ink supported better the structure due to the greater G’. The examples in Table 1 are evidence that both ratios (G”/G’ and G’-G”) should be taken into account when formulating an ink for a medical application, as the printing fidelity could be compromised.

## 3. Cardiovascular Tubular Applications

Several researchers have attempted to 3D print custom cardiovascular applications due to the growing need for tissues and organs for transplantations and research [37,38]. It should be noted that a large percentage of the cardiovascular system is based on tubes, vessels, and arteries. Therefore, the possible solutions to these problems are also cylindrical based, as shown in Figure 1. The main problem when 3D printing cylindrical biomedical devices is that regular 3D printers print on flat surfaces. This fact leads to the use of support material, if horizontal printing. On the other hand, if printed vertically, the 3D printed part does not have uniform mechanical properties, especially to traction [39]. Either way, the medical device is the result of the summation of the deposited layers, so it is not ideal from the force distribution point of view. To overcome that issue, a novel tubular 3D printing approach has emerged in the cardiovascular applications. This technique uses the viscosity of polymers to deposit the materials on a rotating cylindrical mandrel, rather than on a flat surface [40]. This manufacturing approach, combined with the DIW technique, has been used to fabricate biodegradable stents and vascular grafts to avoid layered deposition of the materials.

### 3.1. Vascular Graft

Bridging large arteries and veins with capillaries is a great challenge in terms of tissue engineering, especially because vascular grafts with diameters less than 6 mm evoke early thrombotic occlusion and intimal hyperplasia, caused by the thrombogenicity between the synthetic surface and the native vessel [42,43,44]. This fact is due to the higher area to blood volume ratio [45]. For many years, researchers have attempted to construct a complete human blood vessel with tissue engineering. One of the firsts approaches to build vascular grafts was based on the development of biocompatible mats with adequate biomechanical properties and hemocompatibility. Consecutively, the mats were wrapped in a tubular support to create the vessel shape. Heureux et al. [46] create mats with various cell types by employing ECM with ascorbic acid to culture smooth muscle cells (SMC) and fibroblasts. First, the sheet containing SMC was wrapped in a 4,6 mm cylinder which was then bind with a fibroblasts ECM mat, thus mimicking the media and adventitia layer of the vessels, as in Figure 2. Then, the support was removed, and endothelial cells (EC) were seeded to recreate the lumen. The tissue engineering outcome correctly met the grafting requirements, such as wall thickness, burst pressure, etc. However, the production time was 30 days, thus making it too long to use it in a personalized patient-specific treatment.

To address this issue, DIW has been employed to reduce the production time and to fabricate a vessel engineered to comply with all the grafting requirements. In the early stages, the vascular grafts fabricated by DIW were bioprinted on a flat surface. Agarose based rods containing cells spheroids were deposited layer-by-layer to construct cylindrical vascular grafts. The different rods had different cell types, tailoring the cell deposition layers and recreating the characteristic morphology of the vessel [47]. However, agarose may not be strong enough to withstand blood bursts, not reported in the study. Hence, the same approach was attempted with hyaluronan hydrogels crosslinked with tetrahedral polyethylene glycol tetracrylates. In that case, agarose microfilaments were only used as support material to print the cell-containing microfilaments inks. The hyaluronan hydrogel has been shown to have a higher storage modulus, so it could be a more promising solution for graft applications [48]. This layer-by-layer rod solution containing different cell types to mimic the vessel-like conditions is interesting to manufacture a custom branched vascular network. However, this bioprinting approach requires maturation of the printed tubes. Therefore, the manufacturing time is extended to weeks. Additionally, the uncertain mechanical properties due to the inhomogeneous fusion between the layers makes this technology not yet suited for implementation for patient-specific treatment.

#### 3.1.1. Silk as a Solution for Grafting Arteries

Silk fibroin scaffolds for grafting have been investigated as they show promising results under working conditions [49,50]. Many approaches have been proposed, such as a double-raschel machine [49,51], electrospinning [50], or crafting the graft with a mold [52]. However, these manufacturing methods do not present the ability to rapidly customize the cardiovascular graft for implementation. To overcome these limitations, the tubular DIW method has been explored. It has provided efficient control over morphology, thus providing a technology that enables the rapid construction of patient-specific cardiovascular grafts. Biomaterials, such as silk fibroin, have shown great performance in DIW due to its shear thinning property and suitable viscosity [53]. Silk is a biocompatible material in which mechanical properties can be adjusted depending on the fabrication process and post-treatment techniques [54]. In addition, it can be mixed with other materials, such as heparin or collagen, to meet the anticoagulant requirements to graft small blood vessels (<6 mm) [55] or enhance endothelial cell proliferation on the medical device [56]. For these reasons researchers have developed small vascular grafts using the DIW tubular method. For instance, Lovett et al. [57] deposited high concentrated silk fibroin (25–30%) on a rotating mandrel. Then, the grafts were lyophilized since the study showed that this treatment exhibited greater roughness and porosity that resulted in greater cell proliferation. Vascular grafts were evaluated and assayed in in vivo conditions. Silk showed a less thrombogenic prospective than polytetrafluoroethylene (PTFE), which it has been used for large vascular vessel substitution [58]. Furthermore, silk had higher EC and SMC cell attachment than PTFE. Therefore, the study described evidence for the possibility that cells generate ECM as silk degrades, thus creating a native end vein. Finally, preliminary in vivo assays, 1 month in duration, demonstrated good performance of the graft since no indicators of occlusion, clotting, or ischemia were found. On the other hand, the PTFE grafts exhibited occlusion during the first 24 h [57].

Avoiding thrombogenic events is one of the key features of vascular grafts. That is why attention has been paided to this feature. Dong et al. [59] evaluated the ideal topography of a vascular graft to prevent thrombotic events. They assessed the surface topography of the internal substrate that is in contact with the blood flow to improve the grafting performance. Microfibers, nanofibers, and a smooth surface were evaluated. A self-developed tubular ink 3D printer was employed to fabricate the vascular graft topographies. Electrospinning of 15% Polycaprolactone (PCL), mixed with chloroform and methanol, was carried out on a rotating mandrel using a 21-gauge needle. Electrospinning settings were modified to conduct nanofibers or microfibers. The smooth internal layers were carried out with DIW. Then, the samples were dried for 3 days to remove residual solvent. The study indicated that the smooth surface enhanced hemocompatibility without altering the blood flow and promoted EC adhesion, proliferation, and migration.

Further studies with silk-based vascular grafts were performed to optimize cell colonization by tuning the porosity of the scaffold to accelerate its degradability, resulting in the remodeling of the scaffold in a definitive native vessel. Rodriguez et al. [60] discussed the effect of the silk bioink molecular weight and its concentration on the viscosity, pore size, and mechanical properties. The viscosity of the solution increased with a higher molecular weight of silk and a higher concentration (*w*/*v*%). According to the authors, the viscosity range in which the silk fibroin could be extruded by DIW was between 0.5 Pa·s and 3 Pa·s. Therefore, the assayed bioinks were adjusted in concentration and molecular weight to comply with this constraint. The silk grafts were tested, and a higher concentration of silk fibroin resulted in a higher elastic modulus. In addition, the smaller molecular weights depicted higher load at break and stiffer behaviors but smaller pore diameters. Therefore, the high molecular weight silk was more compliant but had lower suture retention strength. The obtained data demonstrated that the mechanical properties can be adapted by modifying the concentration and molecular weight. This finding is very relevant since cardiovascular applications require a compromise between rigidity and compliance depending on the application to be performed. Furthermore, the study concludes that a medium high molecular weight had the best balance among suture retention and cell infiltration properties.

#### 3.1.2. Cell Deposition to Fabricate Vascular Grafts

As it has been stated, cell adhesion and proliferation in the vascular graft is not only important for its biointegration; it also shows great potential to fabricate a completely native cardiovascular conduit through bioengineering. That is why researchers are trying to develop a vascular graft with the three cell layers of the vascular artery, as each layer has its role in a functioning blood vessel. The intima layer prevents thrombotic events, the media layer provides contractile behavior, and the adventitia supports structural comportment. In general, mimicking an artery by deposition of the cell types in layers to rebuild a cardiovascular channels has greater potential of success of becoming the gold standard than any other concept implemented so far because it not only meets the biomechanics’ constrains but also the biological morphology [54]. That is why, efforts have been made to integrate them into the tubular DIW technology, as shown in Figure 2.

Liu et al. [61] fabricated a bioprinter for the manufacture of artificial vessels, which was later adapted with a multi-nozzle multi-channel deposition system that could control the temperature (2–45 ${}^{\circ }$C) and the applied pressure (0.1–1 MPa). The nozzle precisely controlled these parameters, so it showed good performance in printing thermosensitive materials. Additionally, this novel extrusion system offers a coaxial nozzle that allows the encapsulation of cells with a biomaterial to prevent the cells from being damaged due to the applied pressure [62]. A precise characterization of the printing parameters allows an accurate control of the material properties, such as pore size, which is relevant for cell growth in designed vascular grafts. Then, they developed an optimized theoretical model to predict the thickness of the alginate-gelatin material. Artificial blood vessels of 3.0, 4.3, and 6.9 mm were fabricated using the tubular rotating rod technique. Different concentrations of alginate and gelatin were blended into an ink and deposited on the mandrel. Then, the ink was cross-linked with calcium chloride. The viscosity of the ink increased with increasing concentrations of alginate and gelatin. Finally, the elected solution for the initial investigation was 3% *w*/*w* alginate and 8% *w*/*w* gelatin with a final viscosity of ∼17 Pa·s. The increase in applied pressure was directly proportional to the artificial blood vessel thickness. On the other hand, the nozzle speed was inversely proportional to the wall thickness. Nevertheless, nozzle speeds lower than 0.75 mm/s, exhibited material accumulation. This research paved the grown to pursue a further investigation on the treatment of cardiovascular disease through angiogenesis of living tissue based on alginate-based solutions [63].

Alginate is commonly used for cell culture because it creates an optimal environment for the cells to proliferate [64]. It is well suited for DIW since sodium alginate shows strong shear thinning behavior [23]. Additionally, its stiffness can be improved by crosslinking with calcium ions to achieve gelation and increasing its concentration [22]. However, solutions with a concentration greater than 5% reduce cell viability, as well as proliferation [64]. Despite this constraint, it has been used in tubular DIW applications for multilayered vascular grafts cells that mimic the cardiovascular system. Gao et al. [39] created a microfluidic vascular graft to mimic the behavior of vessels on a chip to better study the blood vessels applications. A sodium-alginate ink solution (4% *w*/*v*) was partially crosslinked to form a gel during deposition. Then, SMCs were deposited on the alginate solution with a 2000 μm nozzle at 1 mL/min, being the inner part of the simulated vessel, and the fibroblast-loaded alginate-based ink was deposited on top with a 510 μm nozzle. The rod was removed, and the entire structure was immersed in CaCl${}_{2}$ to completely crosslink the structure. Finally, a layer of collagen was deposited on the inner side of the vascular graft to promote cell adhesion of the endothelial cells seeded consecutively, therefore building a multilevel channel with 3 layers, such as that represented in Figure 2. They achieved graft with inner diameters of 2, 4, 6, and 8 mm. The study compares a vertically printed vascular graft on a flat surface against the horizontal tubular approach. The printing accuracy of the tubular approach was not only superior but also allowed different shapes and morphologies of the vascular graft. The vascular graft presented an adequate ultimate strength (0.184 MPa) that decreased slightly during the days of incubation in the reactor. The authors confirmed that alginate did not have sufficient mechanical properties to withstand physiological conditions but suggested that this microfluidic device could be used to study the behavior of cells within the cardiovascular environment, for instance, cells in situations that closely resemble the natural working conditions, providing more reliable results to build on.

To address the weak mechanical properties of the alginate, and tackle the DIW with cells at the same time, a multimaterial bioink has been fabricated by Freeman et al. [65] to make fibrinogen printable. A bioactive vascular graft has been constructed by blending fibrinogen with gelatin. Fibrinogen does not have the rheological properties to be printed with the DIW technique (viscosity < 0.5 Pa·s), but it is a favorable biomaterial for vascular tissue engineering [66]. They performed a heat treatment on the gelatin, to reduce its molecular weight, and then it was mixed with fibrinogen. This solution had good printing rheological properties, high viscosity at low shear rate (∼10 Pa·s), and good shear thinning property. On the other hand, gelatin itself has poor spatial resolution when printed via DIW. Therefore, the combination of the two components had a better outcome than the materials alone. Finally, to create a cell-laden bioactive bioink, fibroblasts were mixed with the solution. The solution was printed from a DIW tubular printer at 1 to 7 mL/min through a 24 gauge nozzle. They showed that the duration of the heat treatment on the gelatin affected its shear thinning properties. Longer times (9 h) had better shape fidelity than shorter ones (1 h). Additionally, the heat treatment increased the Young’s modulus. The presence of cells in the material also enhanced the 3D shape fidelity when printing, thus reducing the thickness of the vascular graft considerably. Finally, the cell viability was above 80% in all the tested conditions, so the viability of the printed cells was assessed. In addition, the vascular grafts withstood a blood burst pressure of 52% of the value of the human saphenous vein and exhibited an ultimate strength an order of magnitude greater than the obtained with the alginate-based bioink (∼1.4 MPa). Hence, it demonstrated a great improvement over alginate-based inks and has potential to become a realistic medical device for implantation.

### 3.2. Stent

Stenting is one of the most frequent interventions in cardiovascular surgery. The stent is a temporary solution to a common problem that is usually treated with a permanent medical device. A permanent stent is not an ideal solution as it provides mechanical support to the artery while it heals. Thus, it is no longer necessary after the healing process. Additionally, there are still unsolved problems, such as late in-stent restenosis, permanent vessel caging, and the fact that it does not accommodate to the dynamic movement and growth of the patient [67]. To overcome these issues, the bioresorbable stent has been developed. The bioresorbable stent has the potential to become a patient-specific treatment, since it can be fabricated in a custom shape for each application. Nevertheless, the implantation of personalized patient-specific biomedical devices, such as a cardiovascular stent, will only be feasible if the fabrication of the stent is highly reduced, approximately 20 min [68]. In doing so, researchers are using 3D manufacturing techniques to reduce the fabrication time. FDM was used to construct Polycaprolactone-Polylactic acid (PCL-PLA) stents under 4 min that show good cell proliferation and high geometry precision (85–95%) [69]. The μCLIP printing method has reduced the customized production time to less than 11.5 min. They printed a methacrylated poly(1-12 dodecamethylene citrate) based ink that polymerizes with the incidence of the UV-light. The resulting scaffold had a radial stiffness comparable to metallic stents and a thickness similar to the commercially available polymeric stents [29].

#### 3.2.1. Ink Stents

The DIW technique on a rotating mandrel has the potential to reduce the stent printing time while adding other features to the medical device in terms of composite materials. For instance, Park et al. [70] characterized a sirolimus bio-absorbable drug-coated stent. The stent scaffold was DIW-printed, since PCL was molten in a stainless steel syringe. The paste was deposited by air pressure (600 kPa) through a 250 μm nozzle. The stent was then coated with a solution of PLGA-PEG and sirolimus dissolved in tetrahydrofuran which was sprayed onto the stent. The stent was sterilized and implanted into a male pig for in vivo analysis. The stent showed a continuous release of the drug sirulimus. The in vivo study showed a reduction in neointimal hyperplasia, inflammation, and thrombus formation, even though this group did not take advantage of housing the molten polymer in a syringe barrel to blend an ink with the drug. The polymeric solution could be modified to create composite materials, as reviewed in the discussion section.

#### 3.2.2. Vascular Anastomosis Stent

DIW can provide the technology to develop other types of stent. Vascular anastomosis is a common procedure during reconstructive vascular surgeries or organ transplantation. The practice requires holding the two arteries together while suturing. Thus, a stent that clamps the two sides together would be beneficial to perform this surgery precisely without blood leakage. The resulting stent should hold the arteries together while the surgery is performed but then dissolve quickly. The dissolution must be progressive, without fragile fracture, and with an anti-clotting material. Farzin et al. [28] fabricated a sugar-based stent through DIW to facilitate the suturing of the vascular anastomosis. Dextran food plasticizer (11% (*w*/*w*)) was added to glucose (29% (*w*/*w*)) and sucrose (60% (*w*/*w*)) to make it more ductile and improve compressive strength. They found that the viscosity decreased linearly with temperature, and temperature thinning behavior, which was used to select the range in which the ink was more fluid to be printed (85–90 ${}^{\circ }$C). Eighteen and 20 gauge needles were used to print the ink, which, under these conditions, had a viscosity of 50–70 Pa·s. A 3 mm stent dissolved in 4–6 min in a dynamic solution of phosphate-buffered saline, thus fulfilling the requirements for rapid dissolution. Moreover, no pieces larger than 2 mm were found. So, no particles were generated that could potentially cause a clot and obstruct another artery. In addition, they could control the thickness fiber range between 0.9 to 4 mm by modifying the gauge needle, the printing speed, or the applied pressure. Therefore, a custom stent could be manufactured that meets all the specific requirements of the arteries. The proof of concept of this device with pig valves was then carried out. No leakage was found during the surgeries, and the stent dissolved after 4 min of perfusion. Even though they used the DIW technology, the tubular printing technique was not required for this application since the stent presented sufficient resistance to compression. Furthermore, no trambolic shape designs were needed, so no supports were required to print the stent on a flat surface with a continuous rod.

Moreover, another silk anastomosis stent approach was found in the literature using both DIW and the tubular approach. The tubular cardiovascular devices have been developed by blending silk with other materials that provided new characteristics to the bioink. It is the case of an anastomosis device that has been constructed through printing a multimaterial bioink through DIW. Six percent (*w*/*v*) silk fibroin was mixed with glycerol to form a 80:20 (dry *w*/*w*) silk:glycerol solution. The bioresorbable device aims to avoid the suture of the two vessels to reduce the complexity of the surgery. It consists of two parts, a cylindrical tube with a barb at each end (125% of the outer diameter of the coated rod) and a tubular sheath clip fabricated using the same technique to seal the two arteries. The assembled devices demonstrated a compressive strength comparable to self-expanding metal stents. Plus, it resisted pressure bursts greater than the physiological conditions [71]. The surgery time was reduced to approximately 1 min due to the new silk-based graft approach. Furthermore, a localized drug delivery system could be fabricated by blending the medicament in the ink formulation since the degradation of the material progresses from the luminal surface to the periphery.

## 4. Discussion

DIW has proven to be a suitable solution for printing several very promising biomedical devices due to its ability to customize the extruded material for the printing purposes [6,9,18,38,65]. The only requisite that the bioink must fulfill is the shear thinning and the rapid self-curing ability. These features can be assessed by evaluating the viscosity of the solution and the relationship between the storage modulus and the loss modulus. After meeting these constrains, the materials that can be printed are unlimited, as well as the applications that these materials can develop. DIW is an especially promising technique for cardiovascular bioprinting, since, in cooperation with an adapted rotating bed, continuous tubular medical devices can be created without the need of printing supports [62,69]. The ability to print on a rotating mandrel is due to the implicit viscoelastic property and the shear thinning characteristic of the material, as well as high viscosity when low shear stress is applied. Additionally, it only requires a sufficiently high storage modulus to maintain the shape after deposition and to hold onto the rotating mandrel against the gravitational force when spun. DIW is very convenient for the cardiovascular applications as it facilitates the formation of a very smooth surface of the intima layer. Roughness of the intima layer has been shown to increase the risk of arterial thrombosis [59]. Therefore, it is a key parameter in the manufacturing process of any medical device that works within a vessel. Moreover, DIW is the predominant technology to print living cells with a good printing accuracy and very high cell viability [22,30,72]. Printing a material with living cells is a promising approach to achieve patient-specific or complex tissue engineering constructs, and it can provide the tools to study complex biological systems in a more realistic environment [39]. Today, formulating a material that can be extruded by DIW, maintains the shape of the tubular 3D pattern, allows high cell viability, and is rigid enough to undergo the blood burst remains a major challenge. Additionally, the material must be biocompatible and should comply with the characteristics to allow the proliferation of the cells and maintain their phenotype. Hydrogels have commonly been used for their biocompatible, water-rich environment that simulates the extracellular matrix [26]. However, they lack mechanical strength, which is why some teams have attempted to develop bioinks with other multimaterials [65]. Finally, the cells have to survive the printing process, so physical stimuli, such as stress, voltage, temperature, light radiation, or chemicals, like organic solvents, and crosslinking agents that can damage the cells cannot be included in the printing process [73]. Nevertheless, DIW is a promising technology, since a suitable bioink which enhances cell growth and fulfills the rheological features required for this technique could be formulated. The biggest challenge would be the shear stress the ink undergoes as it passes through the nozzle. The cells should not undergo very high shear stress; otherwise, the bioink’s cell viability is reduced. This issue could be solved by reducing the applied pressure or increasing the nozzle diameter [72]. Further, even cells that have survived the process may not express the phenotype that they would do under natural conditions [74]. Therefore, attention should not only be paid to cell viability but also to gene expression. With this paradigm, tubular cardiovascular biomedical devices have been attempted.

### 4.1. Cardiovascular Graft

This technique has risen much attention in the field of small cardiovascular grafts. Controlling the outer diameter of the rotating mandrel has shown efficient outcomes in constructing vascular grafts to replace small vessels with great printing accuracy. There are two differentiated approaches, as follows.

#### 4.1.1. Printing without Cells

Silk fibroin has been the most common material to address the solution within this approach [57,60,66,75]. It is a biocompatible material that has good mechanical properties than can be tuned with the manufacturing process. In addition, other materials can be blended with the bioink to add characteristics that enhance the outcome of the solution, such as glycerol to add elasticity [71]. Additionally, it has shown good hemocompatibility, since it does not cause thrombogenic events and expresses solid cell colonization. The integration within the vessels is successful as cells adhere to the device and proliferate. Furthermore, the in vivo assays demonstrated reliable results in animal models, as their performance was superior to current applied solutions [57]. For this reason, this approach may have a better chance of becoming an implementable solutions in the recent future, since it has been proved to be mechanically compliant in real work environments, as well as fulfilling the other requirements of the cardiovascular grafts.

#### 4.1.2. Cell-Loaded Bioinks

Constructing a bioengineered native blood vessel may be the final solution to that problem. Teams have attempted to fabricate multi-layered vascular grafts with cells [47,48,65], such as EC in the intima layer, SMC in the media layer, and, finally, fibroblasts at the adventitia, thus mimicking a real artery distribution, as shown in Figure 2. This is a promising approach because it has the potential to deposit autologous cells from a patient in the correct distribution, incubated in a bioreactor, and the cells would build a biocompatible surrogate blood vessel. To address this, research has been conducted, and DIW with multiple bioinks laden with different cell types have been used to fabricate vascular grafts. Usually, hydrogels are utilized due to their high printing precision and high cell viability [27]. However, the current problem with this approach is that they do not support the mechanical restrictions that the working environment requires. Although teams have tried to improve the mechanical properties of hydrogels with composite materials or post-processing techniques, it has not been sufficient to fulfill the working condition constraints [48]. So, even though the approach is really promising, much work remains to be done to make it feasible for its implementation in a human surgery operation.

### 4.2. Bioresorbable Stents from Bioinks

Cardiovascular stents can also be manufactured with this technique. Despite the fact that most of the stents that are made with the tubular 3D printer have been fabricated using FDM [40,69,76], DIW has started to gain attention as it provides the ability to tune the material to add added value to the cardiovascular stent while maintaining mechanical properties. Until now, PLLA has been the most common material in which the bioreserbable stents have been printed [77]. Therefore, FDM was a suitable technique to manufacture this type of medical device since PLLA filaments could be melted and extruded with this technique. However, composite stents are more difficult to fabricate with this manufacturing process since mixing drugs or radiopaque agents in a filament has some limitations in terms of homogenization of the materials, rheological, and thermal properties for printing, etc. [78]. Nevertheless, DIW enables the possibility of mixing multimaterial inks to provide additional features to the solution. Organic solvents can be used to dissolve different polymers to create inks with the properties of both materials and then evaporate the solvent in a fume hood overnight. This can be used to add drugs, such as sirolimus, which showed a reduction of noeintial hyperplasia and low fibirin score in in vivo studies [70]. Adding drugs to a bioabsorbable stent is ideal, as the drug is progressively released as the stent degrades. Since the control of the stent degradation time can be tailored, the drug can be released in a controlled manner. In addition to cell antiproliferatives, it may also be interesting to consider an anticoagulant drug or one that favors the proliferation of endothelial cells in medical equipment. Moreover, manufacturing stents using this method not only allows us to add drugs but also introduce other materials that provide other qualities that can add value to the cardiovascular stent, therefore boosting the performance of the manufactured stent. One example is addressing the transparency conflict of polymeric stents by adding radiopaque agents. This feature is crucial for a stent, since, during surgical operations, it is used for medical personnel to observe the insertion of the stent into the patient by X-rays. However, the addition of materials should be well considered, as the addition of components may induce thick struts. Struts thicker than 150 μm would not be optimal, since thicker struts impede vessel healing [79] and induce turbulent flow, which causes activation of platelets that cause coagulation [80]. Thus, their relative bulkiness could limit its application in small vessels [81].

DIW and the tubular approach offers the possibility of combining these types of bioinks that provide the tools to create rapid patient-specific bioresorbale stents with expanded properties to tackle a problem that has not yet been perfectly solved. Additionally, cardiovascular stents also work within blood vessels. Hence, DIW can also be employed to formulate cell-embedded inks to improve their biocompatibility, biointegration, and enhance the formation of an ECM while degrading the bioresorbable medical device. Therefore, DIW has the potential to design a stent that mimics the composition of arterial cell layers with SMC, EC, and fibroblasts. In Figure 2, a stent containing the multiple layers of a cardiovascular conduit is proposed as a possible solution to build on. This idea of a cell-loaded bioactivated stent could potentially provide a better outcome than current stents.

#### Vascular Anastomosis Stent

Two different approaches for the fabrication of an anastomosis stent were presented in this review. Both were made with different bioinks and extruded by DIW. Nevertheless, despite the different strategies and materials, DIW has been reported to be a valid solution to improve the treatment of cardiovascular pathologies and create medical devices that enhance the possibilities of success of the surgeries.

## 5. Conclusions

This review analyzed the problematic elements of manufacturing cylindrical cardiovascular applications using DIW. The rheological parameters of the extruded bioink were discussed. It was highlighted that, with this technique, two types of medical devices can be fabricated. Cardiovascular grafts for vessel replacement have already been constructed and showed promising results when printed with materials with good mechanical properties. Other attempts with multi-layered cells and hydrogels have also been reported. Despite these promising results of cell deposition and maintaining conformation and viability, work is still required due to not yet complying with physiological conditions. Additionally, DIW provides a new paradigm of materials that can be implemented to fabricate stents that may contribute to the development of a treatment-specific medical procedure. DIW can blend bioinks to provide added value to the final cardiovascular solution. Nevertheless, only a few teams have investigated this promising approach, so there is room for improvement.

## Figures and Tables

**Figure 1 polymers-13-00077-f001:**
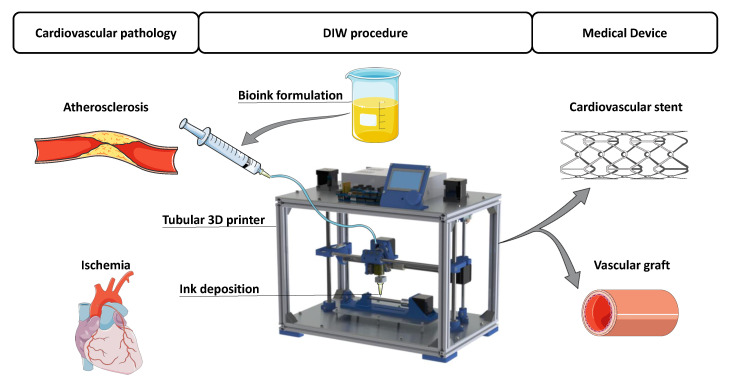
Problems and solutions of the main cardiovascular applications printed with Direct Ink Writing (DIW) [41].

**Figure 2 polymers-13-00077-f002:**
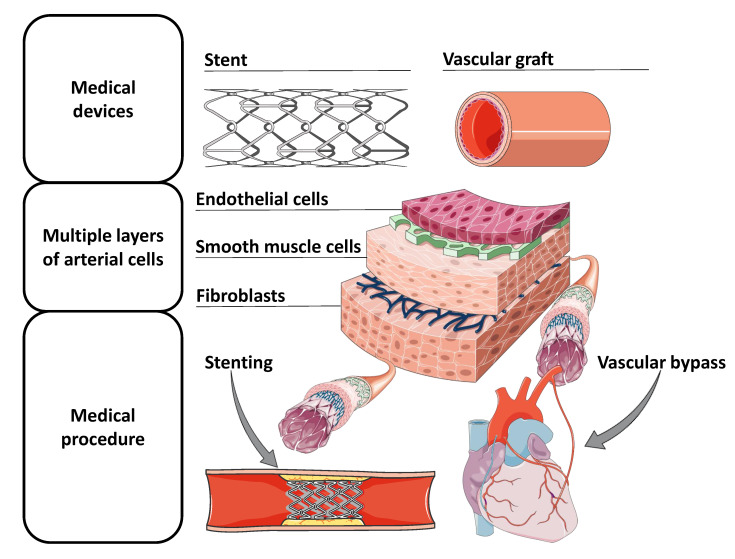
Tubular medical equipment for the treatment of cardiovascular pathologies, representation of the cell layers of arterial vessels, and their implementation within the human body [41].

**Table 1 polymers-13-00077-t001:** Bioink printing fidelity according to the modules G’ and G’’. Print fidelity is a qualitative parameter ranging from 1 to 4. (1) Non-printable. (2) Large scattering. (3) Small scattering. (4) Ink remains where it is placed.

Polymer	Solvent	Application	∼G’-G’’ Difference (Pa)	∼G’’/G’ Ratio	Print Fidelity
Polyacrylamide and 6.5 wt% sodium alginate	Water	Hydraulic Artificial Tentacle [15]	2000	0.5	4
Polyacrylamide and 1.7 wt% sodium alginate	Water	No application [15]	200	0.5	2
poly(methyl-silsesquioxane), SiC and Carbon fibers	Isopropanol	Conductive viscoelastic composite [36]	5000	0.75	3
poly(methyl-silsesquioxane), fumed silica, SiC and Carbon fibers	Isopropanol	Conductive viscoelastic composite [36]	5000	0.38	4
Polymethylsilsesquioxane, fumed silica, ZnO and CaCO3	Isopropanol	Bioceramic scaffolds for bone tissue [14]	2000	0.33	4
Polymethylsilsesquioxane, fumed silica, ZnO, CaCO3 and hardystonite filler	Isopropanol	Bioceramic scaffolds for bone tissue [14]	1300	0.35	4

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
