# Peer review of "Continuous Based Direct Ink Write for Tubular Cardiovascular Medical Devices"

_polymers, 2020, doi:10.3390/polym13010077_

Round 1

Reviewer 1 Report

In this article, the authors analyze the use of Direct Ink Writing technology (DIW) into the novel emerging field of cardiovascular medical application. I have to say that I am little confuse because the article was submitted as scientific paper but it is a review paper. Moreover, as review of an emerging technology such as the DIW is quite poor in terms of contents. In fact, a paragraph that describe the technology is missing and the other related to the bioink used for cardiovascular application have low scientific soundness. In addition, if we look at the bibliography it does not seem suitable for a review article. In my opinion, the article must be completely revised in term of content. Also the way in which the authors divided the paragraphs results confusing and difficult to read.

I would suggest the rejection of the present manuscript on Polymers Journal because in my opinion does not add anything to the present literature and it results not well written and organized.

Author Response

Reviewer #1: In this article, the authors analyze the use of Direct Ink Writing technology (DIW) into the novel emerging field of cardiovascular medical application. I have to say that I am little confuse because the article was submitted as scientific paper but it is a review paper.

The authors lament the confusion and apologize for the inconvenience that this may have caused.

* Moreover, as review of an emerging technology such as the DIW is quite poor in terms of contents. In fact, a paragraph that describe the technology is missing and the other related to the bioink used for cardiovascular application have low scientific soundness.

The paragraph describing the technology has been carefully improved to clarify how this technology works:

On page 2 of the manuscript, the explanatory paragraph can be found:

This technology is based on the extrusion of viscoelastic pastes, also referred as ’inks’, through a nozzle (50 µm to 1.25 mm [6,7]) forming a  3D structure. The shear thinning property in DIW is even more crucial than in FDM, since no heat treatment is conducted, therefore the material relies exclusively on this feature to harden after being deposited on the bed. The shear stress to create the flow is generally generated by the action of a pneumatic or mechanical piston or an Archimedes screw that is connected to the syringe barrel containing the ink [8]. Consecutively, this mechanism is coupled to a nozzle that is mounted on a 3-axis carriage that precisely controls the placement of the deposited material.

In addition, an introductory paragraph of the bioink properties has been added to the document to better introduce the topic and therefore, we hope that the rest of the content would be better understood.

On page 2 of the manuscript:

Designing devices for the cardiovascular system is not an easy task. The blood stream is a very inhospitable environment were any invading body is immediately attacked by the immune system. Therefore, the materials used to design the medical devices that are intended to be introduced in the blood vessels must meet the highest standards of biocompatibility. Additionally, the DIW technique arrives in the cardiovascular field to address differently the issue that has been approached permanently in recent years. Thus, proposing bioresorbable materials that possess sufficiently high mechanical properties to withstand the working conditions without causing blood clots. Moreover, to comply with the DIW requirements, the deposited solution should also fulfill certain rheological characteristics.

* In addition, if we look at the bibliography it does not seem suitable for a review article. In my opinion, the article must be completely revised in term of content.

We have thereby considered the comment on the literature. After going through all the references, the following reference has been discarded:

 Ségry, B. TRABAJO DE FIN DE MÁSTER STENTS POLIMERICOS PARA APLICACIONES 558 CARDIOVASCULARES Memoria Autor. Technical report, 2020.

It is a Master’s thesis work. Hence, it did not undergo under a peer review procedure. Therefore, it lacks scientific relevance.

Furthermore, the document has been analyzed and references have been included where there was a need for scientific support. They can be found in lines 169, 175, 178, 179, 398, 425 and 432.

The references added were as follows:

Jordan, Robert S. and Wang, Yue. 3D printing of conjugated polymers. Journal of Polymer Science Part B: Polymer Physics 2019, 57, 1592–1605. doi:10.1002/polb.24893.

Kiritani, S.; Kaneko, J.; Ito, D.; Morito, M.; Ishizawa, T.; Akamatsu, N.; Tanaka, M.; Iida, T.; Tanaka, T.; Tanaka, R.; Asakura, T.; Arita, J.; Hasegawa, K. Silk fibroin vascular graft: a promising tissue-engineered scaffold material for abdominal venous system replacement. Scientific Reports 2020, 10. doi:10.1038/s41598-020-78020-y

Catto, V.; Farè, S.; Cattaneo, I.; Figliuzzi, M.; Alessandrino, A.; Freddi, G.; Remuzzi, A.; Tanzi, M.C. Small diameter electrospun silk fibroin vascular grafts: Mechanical properties, in vitro biodegradability, and in vivo biocompatibility. Materials Science and Engineering C 2015, 54, 101–111. doi:10.1016/j.msec.2015.05.003.

Moriya, M.; Roschzttardtz, F.; Nakahara, Y.; Saito, H.; Masubuchi, Y.; Asakura, T. Rheological properties of native silk fibroins from domestic and wild silkworms, and flow analysis in each spinneret by a finite element method. Biomacromolecules. American Chemical Society, 2009, Vol. 10, pp. 929–935. doi:10.1021/bm801442g

Ma, X.; Cao, C.B.; Li, J.H.; Zhu, H.S. Novel Prosthesis Using Silk Fibroin for Small Caliber Vascular. Key Engineering Materials 2005, 288-289, 461–464. doi:10.4028/www.scientific.net/kem.288-289.461.

Bosio, V.E.; Brown, J.; Rodriguez, M.J.; Kaplan, D.L. Biodegradable porous silk microtubes for tissue vascularization. Journal of Materials Chemistry B 2017, 5, 1227–1235. doi:10.1039/c6tb02712a.

Zhao, D.; Zhou, R.; Sun, J.; Li, H.; Jin, Y. Experimental study of polymeric stent fabrication using homemade 3D printing system. Polymer Engineering & Science 2019, 59, 1122–1131. doi:10.1002/pen.25091.

Li, C.; Guo, C.; Fitzpatrick, V.; Ibrahim, A.; Zwierstra, M.J.; Hanna, P.; Lechtig, A.; Nazarian, A.; Lin, S.J.; Kaplan, D.L. Design of biodegradable, implantable devices towards clinical translation, 2020. doi:10.1038/s41578-019-0150-z.

*Also the way in which the authors divided the paragraphs results confusing and difficult to read.

The structure and organization of the work has been considered and modified. The following subsections have been created:

3.1.1. Silk as a solution for grafting arteries

3.1.2. Cell deposition to fabricate vascular grafts

3.2.1. Ink stents

3.2.2. Vascular anastomosis stent

4.1.1. Printing without cells

4.1.2. Cell-loaded bioinks

4.2.1. Vascular anastomosis stent

Furthermore, the entire “3.2. Stent” section on page 9 of the manuscript has been edited. As well as the subsequent added subsections. We believe that these changes provide a better structure to the document. We expect that the document is now more appealing to the reader. So less confusing and easy to read. 

* I would suggest the rejection of the present manuscript on Polymers Journal because in my opinion does not add anything to the present literature and it results not well written and organized.

On page 12, section 4.2. Bioresorbable stents from bioinks. New ideas and approaches are introduced for the constructing a bioresorbable stents from bioinks. The authors provide novel ideas, which to our knowledge are not found in the literature, such as blending bioinks with polymers and other materials by dissolving them with organic solvents to provide added value to the DIW-fabricated stent. Different drugs or agents are proposed to improve the current stents. Moreover, we also suggest that DIW can provide the tools to fabricate a bioactive cell-loaded stent, which we not only believe is feasible, but would also provide an innovative treatment for a common problem today.  

Reviewer 2 Report

This is a very well-written review on the application of direct ink writing technique in the fabrication of tubular cardiovascular devices. Overall speaking, the reviewer feels this manuscript has provide in-depth thinking as well as extensive literature search, and therefore hold no objection to the possible acceptance of it for publication in Polymers.

Author Response

Reviewer #2: This is a very well-written review on the application of direct ink writing technique in the fabrication of tubular cardiovascular devices. Overall speaking, the reviewer feels this manuscript has provide in-depth thinking as well as extensive literature search, and therefore hold no objection to the possible acceptance of it for publication in Polymers.

The authors show gratitude for the positive comments of the reviewer and appreciate the consideration for the acceptance of our review in Polymers.

Reviewer 3 Report

I think the paper needs to be checked thoroughly for typos and incorrect wording. For examples, in the Rheological Properties section I think "loose modulus" is supposed to be "loss modulus" and complex module should be "complex modulus". The same mistake happens again on line 344.

Anastomosis devices are disscussed in the paragraph starting on line 164, but then another anastomosis device is discussed in detail in a very different section of the paper (line 257). This was confusing and maybe the topics should be discussed in the same place.

Are all of the figures, including the individual components of each figure original artwork that has never been published before? The parts of the images look stylistically different, like they may have come from different original sources. If that is the case, you need to make sure that you have permission to reprint them and then cite the original source in the paper.

Author Response

Reviewer #3: I think the paper needs to be checked thoroughly for typos and incorrect wording. For examples, in the Rheological Properties section I think "loose modulus" is supposed to be "loss modulus" and complex module should be "complex modulus". The same mistake happens again on line 344.

The authors value positive comments to reinforce the purpose of the work and welcome the interest in improving the manuscript contents. We agree that these were typos, so they were modified to the correct word “loss” or “modulus” on lines 60, 61, 69, 80, 344 and 451. Additionally, the document was extensively revised for other typos or poor English expressions that were modified throughout the document. Finally, if further improvement is required, we would like to request an extended deadline to send it to a professional English editor to review the document.

* Anastomosis devices are disscussed in the paragraph starting on line 164, but then another anastomosis device is discussed in detail in a very different section of the paper (line 257). This was confusing and maybe the topics should be discussed in the same place.

We acknowledge that discussing this topic in two different sections can be confusing. Therefore, for the sake of a good understanding of the review and to convey the ideas well, a new paragraph has been included to exclusively address this topic [line 318 – 353].

* Are all of the figures, including the individual components of each figure original artwork that has never been published before? The parts of the images look stylistically different, like they may have come from different original sources. If that is the case, you need to make sure that you have permission to reprint them and then cite the original source in the paper.

We would like to clarify that the composition of the images has been made by the authors. The images come from two different sources. The tubular 3D printed image comes from Albert Roca’s undergraduate thesis project, https://dugi-doc.udg.edu/handle/10256/14561. The intellectual property policy of the University of Girona establishes that all the work conducted within the university with its resources is part of the university or research group where it has been developed. He carried out the project under the supervision and resources of our research group. Therefore, we can use the image for this figure without any objection.

The second source is an open site that offers free medical images (https://smart.servier.com/). The authors thought that the images were totally open for use and exploitation. However, after reading the terms of use, it has been noticed that without authorization they cannot be used for publication. We have contacted them to obtain their authorization and we will be back to you after their response.

Round 2

Reviewer 1 Report

The authors strongly improved the quality of the manuscript after the review process. As requested by the reviewer they re-organize all the manuscript by modifying the structure of the work. Many paragraphs have been added and now the manuscript is more comprehensible. In my opinion, the manuscript appears as an exhaustive review of the DIW technique with a special focus on cardiovascular medical devices.

In addition, the bibliography have been improved by adding many works on this topic. I would suggest the acceptance of the present manuscript in Polymers journal.